# Grain-Size Distribution of Surface Sediments in the Chanthaburi Coast, Thailand and Implications for the Sedimentary Dynamic Environment

**Chengtao Wang [1], Min Chen [1], Hongshuai Qi [1,2,*], Wichien Intasen [3] and Apichai Kanchanapant [3]**

[1]   Institute of Oceanography, Ministry of Natural Resources, Xiamen 361005, China
[2]   Fujian Provincial Key Laboratory of Marine Ecological Conservation and Restoration, Xiamen 361005, China
[3]   Department of Mineral Resources, Ministry of Natural Resources and Environment,
     Ratchathewi Bangkok 10400, Thailand
*   Correspondence: qihongshuai@tio.org.cn; Tel.: +86-0592-219-5354

**Abstract:** This paper analyzes the grain-size distribution of surface sediments of the Chanthaburi coast of Thailand to investigate the sedimentary environment and its evolution to better use and protect the coast. The Flemming triangle method, the grade-standard deviation method, and the Gao–Collins grain-size trend analysis method (GSTA model) were used to study the dynamic sedimentary environment of the area and provide preliminary identification of source materials. There are seven types of surface sediments on this coast, with grain sizes ($\varphi$) generally consisting of sand and silt. Sorting is generally poor, and becomes gradually poorer with distance offshore. Skewness is generally positive. The study area is mainly composed of sand and silt, indicating that the hydrodynamics are strong. The results of grade-standard deviation analysis indicate that sediment grain size b (3.25–4.5$\varphi$) is a sensitive indicator of environmental change. This sediment type exhibits a relatively complex transport trend, mainly characterized by northwestward and northeastward transport from sea to land. Sediments at the mouth of the Chanthaburi Estuary and the Welu River fluctuate under the influence of tidal currents. Based on the results of grade-standard deviation analysis and grain-size trend analysis, the study area was divided into three provinces, representing different sedimentary environments and material sources. Compared with tidal-controlled estuaries in the temperate regions of eastern China, the two tropical estuaries examined in this study exhibited smaller suspended sediment loads, runoff amounts, and tidal ranges. However, hydrodynamic conditions were generally stronger. The main reasons for the similarities and differences in the transport trends of sediments in these estuaries were differences in hydrodynamic conditions and the specifics of regional topography.

**Keywords:** surface sediments; grain-size transport trend; grade-standard deviation method; sedimentary dynamic environment; Gulf of Thailand; Chanthaburi coast

## 1. Introduction

The Gulf of Thailand is located on the Sunda Shelf of the South China Sea. Water depths are generally shallow, and the gulf is largely surrounded by the land masses of Thailand, Cambodia, and Malaysia. The Gulf of Thailand covers an area of about 35,000 km$^2$. Most of it has a tropical monsoon climate. Currents in the gulf are affected by the monsoon and change with the seasons [1]. The study area is located on the east coast of Thailand. The tidal current is associated with a regular diurnal tide with an average tidal range of 0.8–1.2 m. Coastal currents in this area are affected by the southwest monsoon in summer, which results in a southeast coastal current, whereas in winter, the area is affected by the northeast monsoon, forming a northwest coastal current [2]. Seasonal variations

play an important role of water circulation patterns in the Gulf of Thailand [3]. Human activity is high within the study area, including relatively high levels of economic development along the coast. By analyzing the surface sediment characteristics of the Chanthaburi coast of Thailand, this study provides a scientific basis for further revealing the evolution of the sedimentary environment, contributing to the utilization and protection of the coast. The results of this study also include fundamental datasets necessary to conduct further studies of the interactions between human activities and natural processes of the coastal zone in this region.

Sediment grain-size data provide extensive information on sediment transport and sedimentation and are fundamental to understanding the hydrodynamic characteristics of the corresponding sedimentary regions, as well as the associated sediment transport and deposition processes [4]. The contributions of distal and proximal sediment sources are preserved in different grain sizes, the variables of which are strongly related to aggradation/degradation processes in alluvial and coastal areas [5]. Over the last two decades, the concept of sensitive grain-size components has emerged, referring to the variation of size fractions with changes in hydrodynamic intensity. Current methods to identify the sensitive grain-size component include the grade-standard deviation method [6,7], the factor analysis method [8], and the end-member modeling analysis method [9,10]. Using the grade-standard deviation method, previous studies have successfully identified the sensitive grain-size component to indicate changes in the dynamics of the sedimentary environment [6]. Further analyses such as grain-size trend analysis (GSTA), which evaluates the mean size, sorting coefficient, and skewness of the sediments, can be carried out to additionally characterize grain-size characteristics and provide insight into the sedimentary environment [11,12]. The more recently developed Gao–Collins method improves upon GSTA by implementing two-dimensional grain-size trend analysis, whereby changes in grain-size parameters are identified in a planar distribution [13,14]. This method has been applied to many marine environments including the Bohai Sea, the South China Sea, and the Mediterranean Sea [15–17]. The combination of the GSTA and the grade-standard deviation method can not only explain the dynamic process of grain-size distribution, but can also determine the transport trend of grain size. For this study the distributions of grain-size parameters and sediment types were examined to analyze the sedimentary dynamic environment and trends in sediment migration in the Gulf of Thailand off the Chanthaburi coast of Thailand. Methods included the Flemming triangle method, the grade-standard deviation method, and the Gao–Collins two-dimensional grain-size trend analysis method.

## 2. Materials and Methods

### 2.1. Sediment Sampling

Samples were obtained from the Third Institute of Oceanography, Ministry of Natural Resources and the Mineral Resources Bureau of the Ministry of Natural Resources and Environment of Thailand, and were originally collected in November 2015 using a cylindrical box sampler on the Chanthaburi coast of Thailand. Surface sediment samples were collected from a total of 94 stations (Figure 1). Surface samples taken from the top 0 to 5 cm at each station were selected for grain-size analysis.

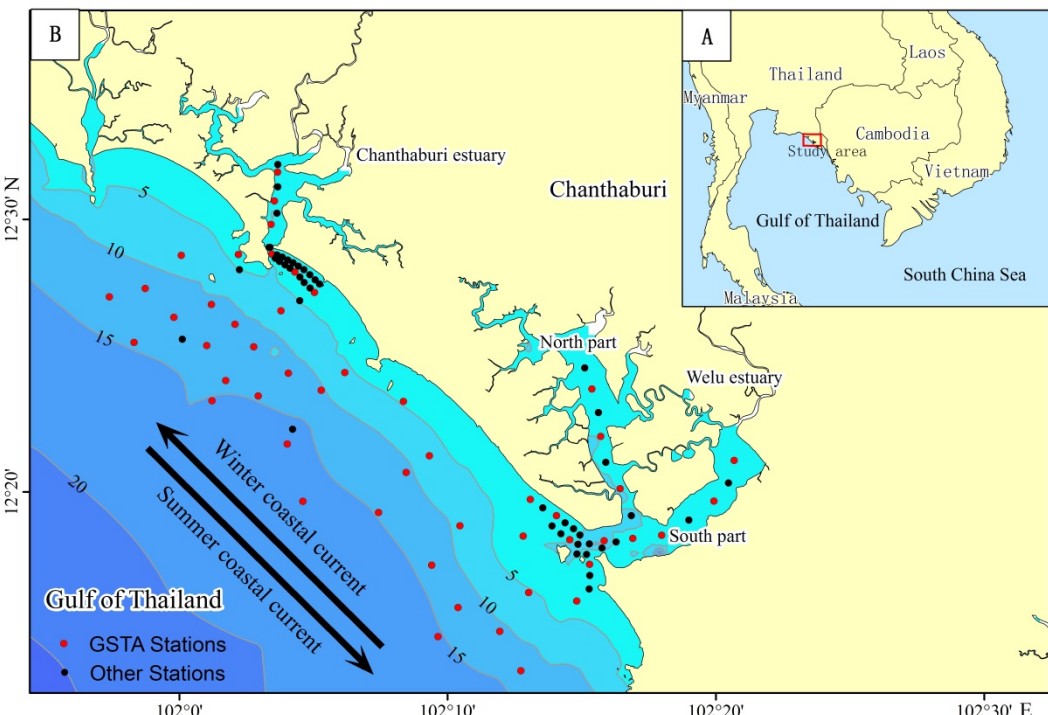

**Figure 1.** Maps showing (**A**) the location of the study area and (**B**) surface sample sites (GSTA Stations represent the grain-size trend analysis stations, and Other Stations represent the stations where grain-size trend analysis was not performed), direction of coastal current from [1,18].

## 2.2. Sediment Grain Size Measurements

The collected surface sediment samples were subjected to grain-size analysis. For each sample, approximately 1 g of wet sediment was weighed. Hydrogen peroxide was added to remove organic matter. Then, 5 mL of dilute hydrochloric acid was added to decalcify the sediment. After acid washing, hexametaphosphate (0.5 mol/L) was added. The particles were completely dispersed after 24 h. Then, the sample was measured using a Mastersizer 2000 laser grain-size analyzer. The test range was from 0.02 to 2000 μm, and the relative error of the instrument was within 3% [19–21].

## 2.3. Grain-Size Transport Trend Analysis

The first step in grain-size trend analysis is to compare every two adjacent sample points on a sample point grid to find all grain-size trend vectors [22,23]. In marine environments, there are many possible variations between any adjacent sampling points A and B. For example, the mean size ($\mu$), sorting coefficient ($\delta$), and skew coefficient (*Sk*) of samples taken at point A might be alternately larger or smaller than those of samples taken at point B. Each combination of these can be represented by a vector. For example, type 1: $\delta_A < \delta_B$, $\mu_A < \mu_B$, $Sk_A > Sk_B$, type 2: $\delta_A < \delta_B$, $\mu_A > \mu_B$, $Sk_A < Sk_B$, etc.; any one of the grain size trends can be represented by a vector [24]. Whether the two sampling points are "adjacent" can be measured by the characteristic distance $D_{cr}$ ($D_{cr}$ is usually the maximum sampling interval). If the actual distance between the two sampling points is less than $D_{cr}$, it is judged as "adjacent." Otherwise, it is judged as "not adjacent." The second step is to find the sum of the trend vectors for each sample point. Finally, a surface vector distribution of the surface sediment grain size can be obtained [22,23,25,26]. Gao encoded the Fortran program GSTA that can be used for surface sediment transport trend analysis [14].

## 3. Results

### 3.1. Sediment Distribution Characteristics

According to the results of grain-size analysis, the classification and naming of sediments in the study area were determined according to the Shepard triangle classification method [27]. There were mainly seven types of surface sediments in the area (Figure 2): silt (T), sandy silt (ST), silty sand (TS), fine medium sand (FMS), gravelly sand (GS), gravel-sand-silt (GST), and clayey silt (YT). Among them, silt (T), sandy silt (ST), and silty sand (TS) were the most widely distributed sediments, and were distributed throughout the vast majority of the study area. Fine-grained silt (T) associated with low hydrodynamic strength was primarily distributed in the offshore area. Medium-grained sandy silt (ST) and silty sand (TS) reflect stronger hydrodynamic potential of the sediment. These fractions were widely distributed in the study area, but concentrated in the northern part of the study area, the far shore area, and the southern part of the Welu Estuary. The other types of sediments exhibited limited spatial distribution, generally occurring as distinct blocks or patches.

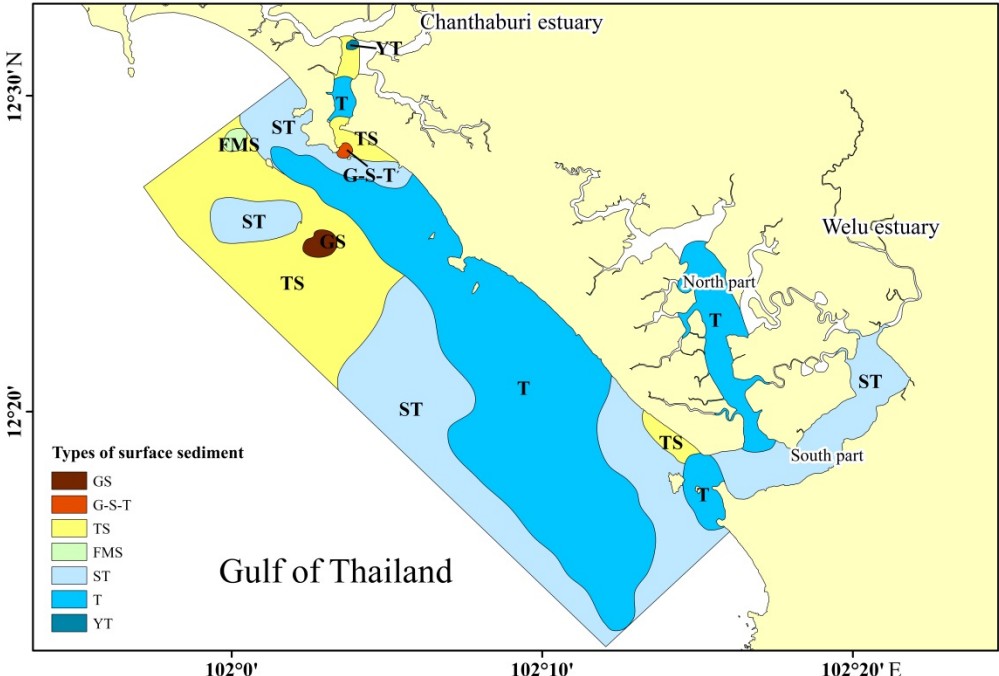

**Figure 2.** Surface sediment distribution: silt (T), sandy silt (ST), silty sand (TS), fine medium sand (FMS), gravelly sand (GS), gravel-sand-silt (GST), and clayey silt (YT).

### 3.2. Planar Distribution Characteristics of Sedimentary Grain Groups

Surface sediments are generally composites of various grain sizes. Thus, we divided sediments in the study area into three classes: sand, silt, and clay. The characteristics of sedimentary environments within the study area could then be determined based on the planar distribution of the three grain-size fractions. As presented in Figure 3, silt comprised the largest fraction in surface sediments, followed by sand and clay.

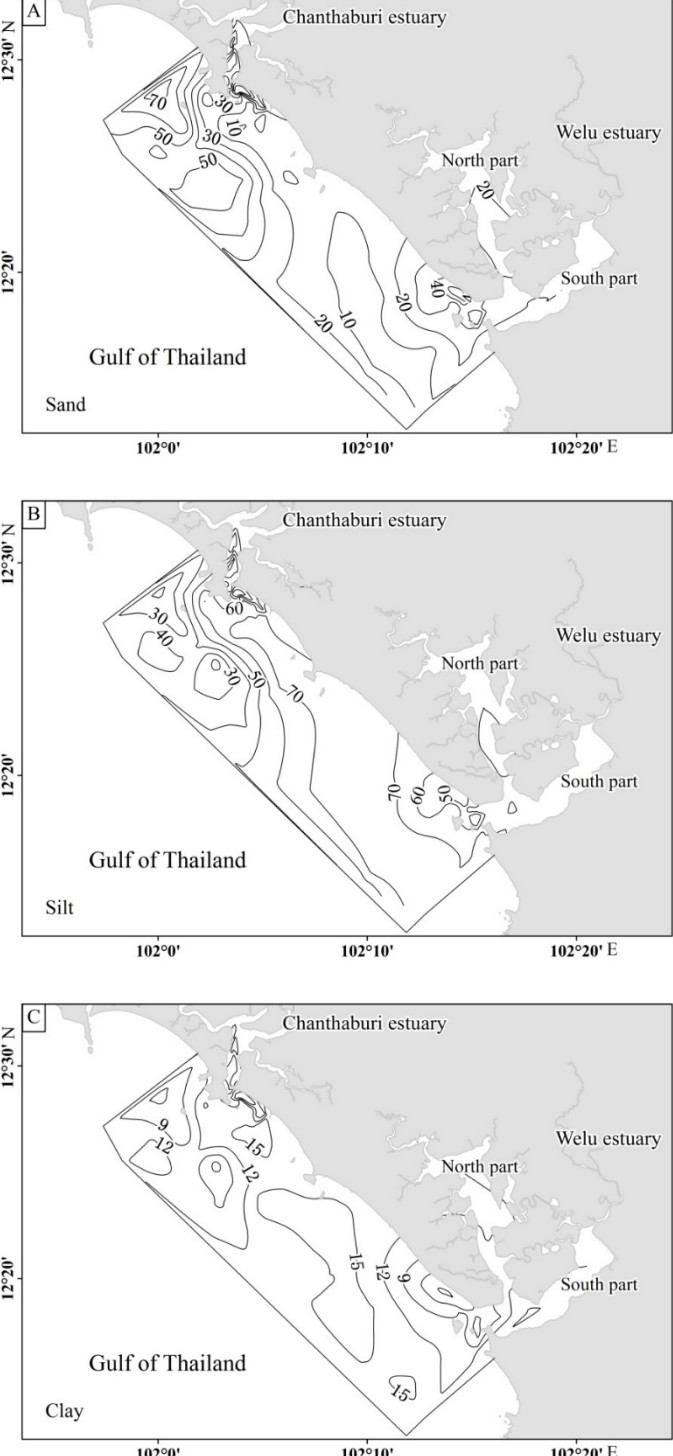

**Figure 3.** Contour maps of the sand (**A**), silt (**B**), and clay (**C**) volume fractions.

Sand grains are coarser and move primarily by shifting on way. Areas with high sand content were primarily distributed southwest of the Chanthaburi Estuary. Additionally, there were two small areas with high sand contents where the sand fraction was greater than 50%. The volume fraction of sand in the Chanthaburi Estuary was more complex. The sand content at the mouth was higher, and the outward extension showed gradually reduction in sand content. Under the influence of the tidal current, the sand volume fraction gradually declined with distance offshore. In the Welu Estuary, the

sand content first decreased and then increased with distance offshore. The coarser sand and gravel brought by the river was deposited within the estuary.

Fine-grained silt is generally transported in a suspended form. At the mouths of the Chanthaburi and Welu estuaries, silt is transported outward. The silt contents of these two areas tended to first increase and then decrease with distance from the river mouth. In the central coastal area between the two estuaries, the water depth gradually decreased from sea to shore, and silt was easily transported by tidal currents to the coast. The silt content of coastal areas reached a maximum of approximately 70%, and silt content generally increased from sea to land. The distributions of the silt and sand fractions were negatively correlated throughout the study area.

Clay is the finest sediment fraction. As a result, clay can be transported farther than silt. Reflecting the strong hydrodynamic conditions in the area, the clay fraction was low throughout the study area, and the maximum value did not exceed 20%. In the northern part of the study area, the shore component extended to the sea. The clay component first increased and then decreased, and two low-value areas appeared in the area. The content of clay components was highest in the central coastal area between the two estuaries. In the south of the study area, clay content gradually increased with offshore distance. The clay fraction exhibited a strong positive correlation with the silt fraction. Clay grains are finer and more easily transported by the tide, as well as more easily transported offshore.

### 3.3. Analysis of Sediment Grain-Size Characteristics

Sediment grain-size characteristics reveal a great deal about the sedimentary environment and can be used to identify sediment sources [28,29]. The mean size represents a concentrated tendency of the grain-size distribution and depends to some extent on the grain-size distribution of the source material. The mean size ($\mu$ value) of the study area was between 1.5 and 6.7$\varphi$ (Figure 4). The $\mu < 4\varphi$ contour reflects coarser sand-based sediments, mainly distributed in the southwestern part of the Chanthaburi Estuary. The contours of $\mu$ values in the interval 4–6$\varphi$ reflect the distribution of coarse silt, which was mainly distributed in the coastal waters, including most of the central and southern parts of the study area. Overall, the sediment grain sizes in the northern part of the study area were coarser, and the sediments in the central and southern areas were slightly finer.

The sorting coefficient is a good indicator of sediment sorting, indicating the uniformity of grain sizes. It is often used as an environmental indicator to represent the relative sedimentary environment. The sorting coefficients ($\delta$ values) of the study area were between 0.79 and 4.10, and the average value was 2.08. The sorting coefficient was poor. In most areas, the sorting coefficient was greater than 2, and the sorting coefficient value was less than 2 only at the mouths of the two estuaries. The sorting coefficients in the northern and central parts of the study area were greater than 2, which represented poor sorting. The sorting coefficient gradually increased with distance offshore, indicating that sorting gradually became poorer. The sorting coefficients near the Welu Estuary were generally between 1 and 2, indicating poor sorting. As shown in Figure 4, the sorting coefficients of surface sediments in the study area were generally poor. The sorting gradually became poorer from the Welu Estuary and coastal areas to the central and northern Chanthaburi Estuary.

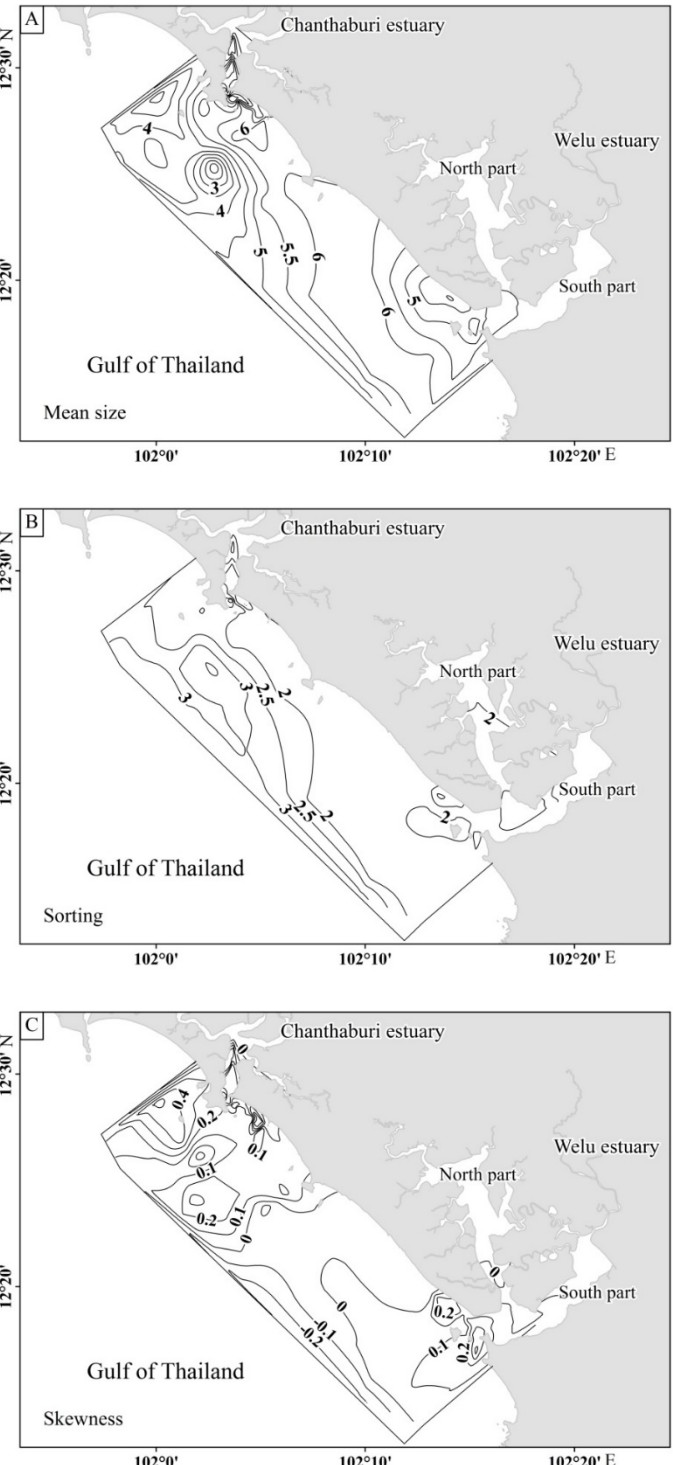

**Figure 4.** Contour maps of mean size (**A**), sorting coefficient (**B**), and skewness (**C**).

Skewness can be used to discriminate the symmetry of the distribution, which can be described by the relative positions of the mean, median, and mode. If the skewness exhibits a negative bias, the sediment is biased toward coarse grain fractions, whereas a positive skewness indicates a bias toward fine fractions. Skewness can also be used to identify the cause of sedimentation. As shown in Figure 4, the skewness values (*Sk*) of the study area ranged between −0.29 and 0.60, with an average of

0.14. Most areas were positively biased, reflecting slightly greater prevalence of fine sediment than coarse sediment.

### 3.4. Sensitive Grain-Size Component

The sensitive grain-size component of the sediment can be determined based on the grade-standard deviation method [6]. Using this method, two sensitive grain-size component intervals are identified and designated as sensitive grain size a (1.25–2$\varphi$) and sensitive grain size b (3.25–4.5$\varphi$). Comparing these two size fractions, the grade-standard deviation value of sensitive grain size b is larger, indicating that this grain size is more prone to fluctuation within the study area and likely to be a better indicator of environmental change. Therefore, sensitive grain size b was selected as an environmental indicator for the study area. The percentage of sensitive grain size b at each station was summed and plotted according to the percentage of the sensitive grain size, as shown in Figure 5. Sediment samples containing high percentages of sensitive grain size b were located near the east and west sides of the mouth of the Chanthaburi Estuary and near the mouth of the Welu Estuary, whereas samples from other areas contained relatively little of this size fraction.

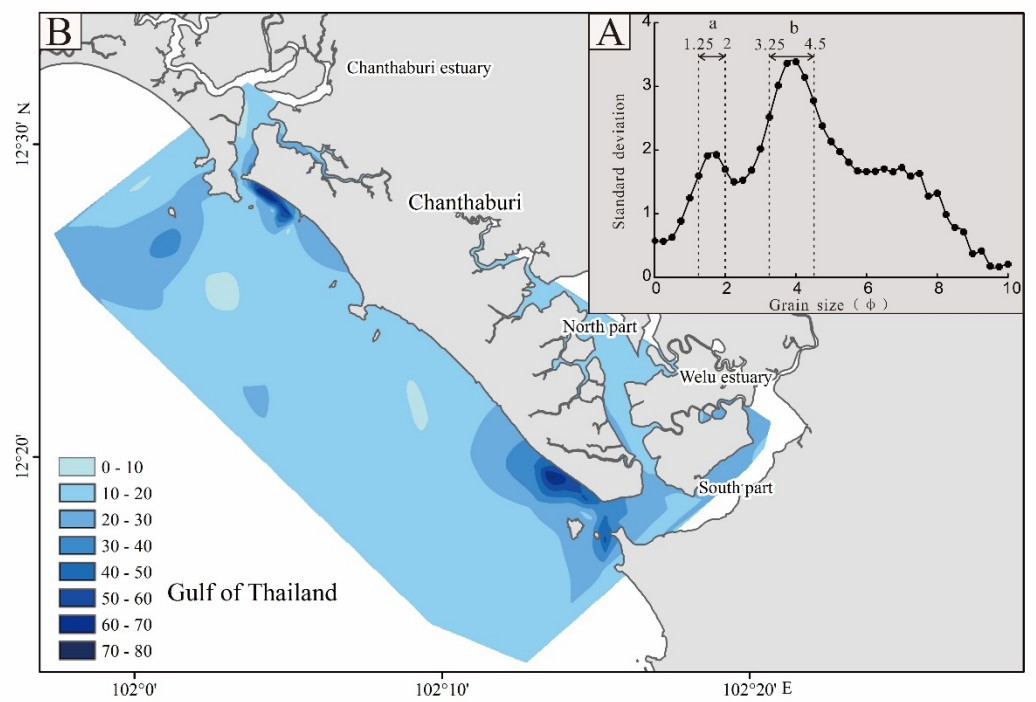

**Figure 5.** Map showing (**A**) the grade-standard deviation curve and (**B**) the distribution of sensitive grain size b contents.

### 3.5. Surface Sedimentary Dynamic Division

In the Flemming triangle, there are 25 zones that indicate different sedimentary environments [30]. According to the volume fractions of sand in the sediments, those sediments are divided into six components, with 95%, 75%, 50%, 25%, and 5% as structural classification lines. According to the volume fraction of clay, sediments are divided into six different hydrodynamic zones (I–VI), with 10%, 25%, 50%, 75% and 90% as the structural classification lines. From I to VI, closer to the clay end, the hydrodynamics are weaker. The results of grain-size analyses conducted as part of this study were projected in the Flemming triangle (Figure 6). The grain-size compositions of the sediments in the study area were mostly silt and sand. They were mainly distributed in Zone I and Zone II and far from the clay end, indicating that the hydrodynamics of the study area were strong.

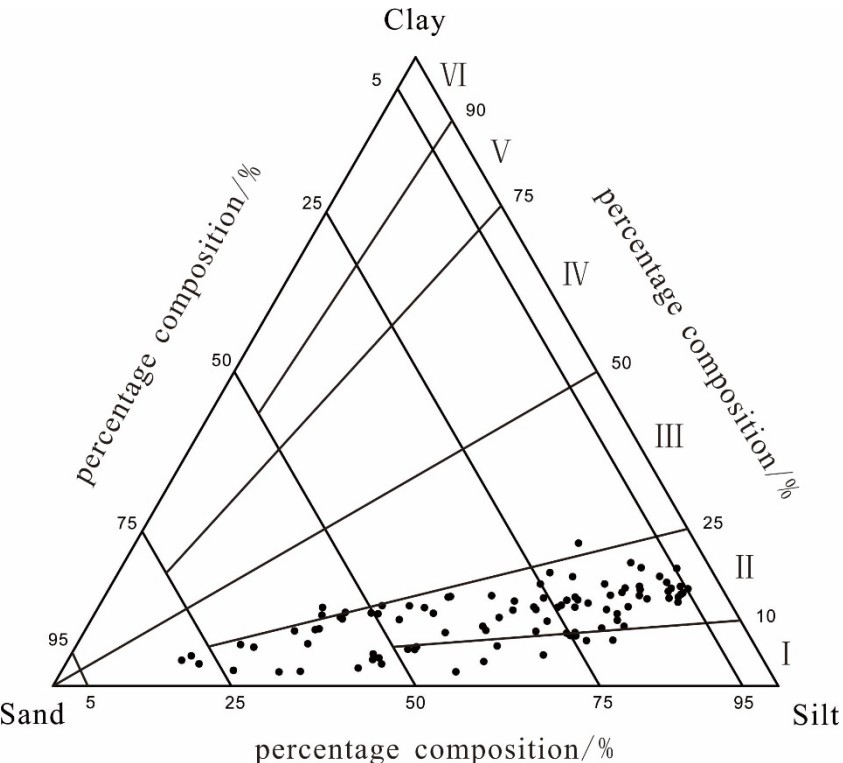

**Figure 6.** Sedimentary dynamic division triangle, based on method from [30].

### 3.6. Sediment Grain-Size Trend Analysis

The grain-size trend refers to the planar distribution of changes and trends in sediment grain-size parameters [24]. Gao proposed a two-dimensional "grain-size trend analysis" method to calculate the combined vector of each sample point to obtain the grain-size trend of the point [23]. In this study, before software analysis was conducted, the GSTA model was used to perform grid pretreatment on the grain-size data of samples collected along the Chanthaburi coast of Thailand. The denser estuary area in the sampling station was removed, and finally the grain-size trend analysis was performed on the remaining 50 stations. Through the use of multiple search radii, multiple calculations, and cartography in the GSTA model Fortran program, the radius of 4 km was found to truly reflect the trend of the sediment. Then, a net transport trend map for sediments (Figure 7) was obtained in which the vector arrows represent the net transport direction of the sediment.

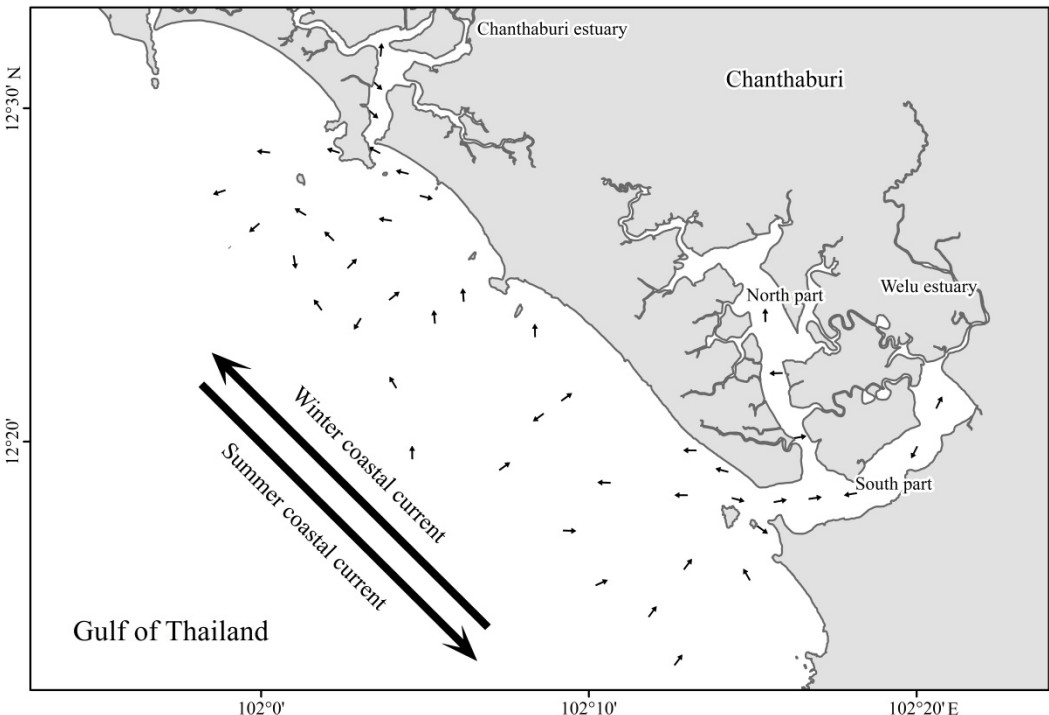

**Figure 7.** Spatial distribution of the net transport vectors of surface sediments.

## 4. Discussion

### 4.1. Division of Sedimentary Environments and Preliminary Analysis of Sediment Provenance

The sensitive grain-size component responds sensitively to changes in the sedimentary environment, and change in the content of this fraction represented the main factor affecting differences in the sediment grain-size distribution within the study area. Changes in the sensitive grain-size component can be used as a sensitive substitution index for regional sedimentary dynamics and changes in the sedimentary environment [31,32]. Combined with the results of grain-size trend analysis, the modern sedimentary environment of the Chanthaburi coast within the study area could be divided into three provinces (Figure 8). Chen et al. [33] analyzed surface diatoms in the coastal area of Chanthaburi, and identified four diatom assemblages representing different environmental conditions. The environmental conditions indicated by these diatoms were linked with the sedimentary environments indicated in this paper.

Province I: Province I is distributed in the nearshore region of the Chanthaburi and Welu estuaries and is characterized by a high fraction of sensitive grain size b. Sediment grains are mixed, consisting mainly of sand and silt. The sediment types are mainly silty sand and sandy silt. The hydrodynamics of the sediments in this area were the strongest. The area was obviously modified by the terrain. The percentage of sensitive grain size b was significantly higher outside the mouth and at the mouth, and high values were primarily distributed in the nearshore area. The sediments located in the mouth of the area exhibited a reciprocating bidirectional transport trend, which further indicated that the hydrodynamic changes in the area were significant. The diatoms in the area were dominated by saltwater species and also contained some freshwater species, which indicates that the interactions of forces between runoff, tidal currents, and waves were significant [33]. This finding was consistent with the conclusions of the sensitive grain-size component analysis method. The sediments brought by runoff were in an environment of strong dynamic changes; the coarser sediments were deposited near the mouth of the area, whereas finer sediments were transported farther offshore. Based on the GSTA analysis results, the surface sediments in this area were mainly terrigenous sediments brought by runoff, mixed with a small amount of sediment from the surrounding sea areas.

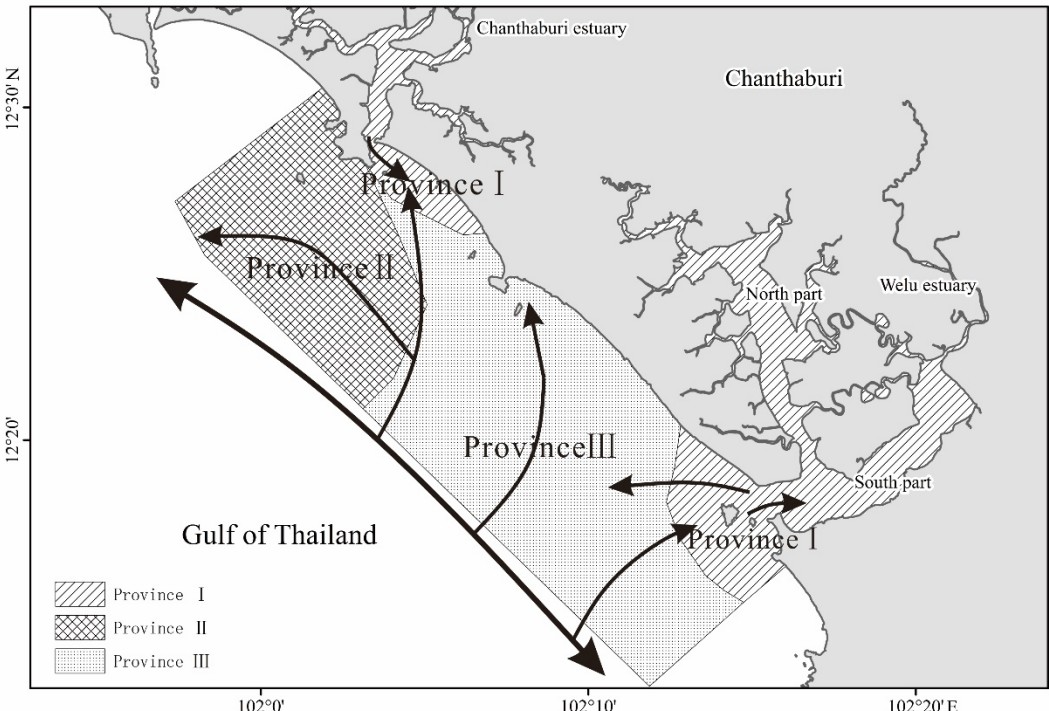

**Figure 8.** Transport trends of sediment grain size and division of sedimentary environments.

Province II: Province II is distributed in the northwestern part of the study area. The fraction of sensitive grain size b is high. Sediments are coarse-grained and the erosion effect is significant. The sediment is mainly sand, and there are two areas with high sand contents where the sand fraction is greater than 50%. Some mixed gravel deposits also occur. The sediment grain sizes of this area were the coarsest in the whole study area, indicating that it has the strongest hydrodynamics. The area is also characterized by strong hydrodynamic changes. Diatoms in this area included more warm-water species, which represent the water body of the Gulf of Thailand [33]. Under the erosion of strong tidal currents and continuous coastal currents, fine-grained sediments at the surface in this area were gradually removed from the study area, leaving coarser sediments of fine sand and coarse silt. Combined with the westward transport trend of surface sediments determined by GSTA analysis, the surface sediments in this area are relatively coarse sediments remaining in the denuded state.

Province III: Province III is distributed in the central part of the study area as well as most of the coastal areas, and is characterized by low fractions of sensitive grain size b. The sediment is fine-grained, consisting mainly of silt. The regional hydrodynamic changes are not obvious. Province III is primarily influenced by tidal currents, which bring considerable amounts of fine-grained sediments from the open sea. With decreasing offshore distance, the surface sediments are generally finer, and the fraction of fine-grained sediments in the coastal areas is as high as 80–90%. According to the study of Chen et al. [33], the diatom assemblage in this area comprises mainly warm-water coastal species, with common warm-water species found in the South China Sea. Based on GSTA analysis, the transport trends of surface sediments were more complex in this area, but generally exhibited northwestward- and northeastward-trending transport from sea to land. This suggests that the fine-grained sediment in the area is material from the Gulf of Thailand brought by tidal currents and other processes.

*4.2. Comparison with Tidal-Controlled Estuaries in the Temperate Region of Eastern China*

Chinese tidal-controlled estuaries are primarily distributed in the southeastern region of the country, including the Oujiang Estuary in Zhejiang Province, the Jiulongjiang Estuary in Fujian Province, and the Pearl River Estuary in Guangdong Province [34]. Most of these tidal-controlled

estuaries are classified as strong tidal estuaries, and the sediments are generally dominated by marine sediments, supplemented by terrigenous sediments. However, the two estuaries in the study area are classified as tropical estuaries. The sampling field campaign took place in November, during the dry season of the winter monsoon, with low rainfall and low river sediment input. Thus, most of the sediment transport was conveyed by tidal movement. Saline density currents and corresponding distal sediment transport were not observed in this location [35]. The suspended sediment content and runoff of the two rivers are low, and the average tidal range is small (0.8–1.2 m). In contrast, several of the Chinese river estuaries have large suspended sediment contents, runoff amounts, and tidal ranges (Table 1). The study area is similar to several estuaries in China, and both areas have obvious wet and dry seasons. The suspended sediment concentration is mainly concentrated in the wet season (May to August), whereas the dry season (November to January) is characterized by lower suspended sediment concentrations [36,37].

**Table 1.** Main river information.

| River | Climate Zone | Nation | Tidal Range (m) | Suspended Sediment Concentration (kg/m$^3$) | Runoff (m$^3$/Year) |
|---|---|---|---|---|---|
| Chanthaburi | Tropical | Thailand | 0.8–1.2 | 0.023 | - |
| Welu | Tropical | Thailand | 0.8–1.2 | 0.016 | - |
| Oujiang [38,39] | Temperate | China | >4 | 0.131 | $469.1 \times 10^8$ |
| Jiulongjiang [40,41] | Temperate | China | 4 | 0.21–0.23 | $148.05 \times 10^8$ |
| The Pearl River [42–44] | Temperate | China | 1–1.7 | 0.1–0.3 | $1741 \times 10^8$ |

Sediments in the Oujiang Estuary are derived from both marine and terrestrial sources. The two types of sediments differ significantly in terms of grain-size parameters and mineral characteristics, providing obvious clues as to the origin of these sediments. The clay mineral composition of the suspended sediment in the Oujiang Estuary is completely different from that of soil, suggesting that it is primarily derived from a marine source [45]. The tidal current exhibits a reciprocating motion along the deep trough of the channel [39]. The fine-grained sediments in the Chanthaburi and Welu estuaries are likewise predominantly of marine origin, which means that the fine-grained sediments in the three estuaries are similar. The sediments in the Jiulongjiang Estuary are primarily clayey silt, but are complex, and additionally affected by summer typhoons. Compared with samples collected in the 1980s, the sediment grain-size in the western part of the estuary shows a coarsening tendency and is formed under the erosion of strong hydrodynamic conditions [46]. This is similar to the composition of coarse-grained sediment in the southwestern part of the Chanthaburi Estuary in the study area, which is preserved after strong hydrodynamic erosion. The sediment transport trend in the surface sediments of the Jiulongjiang Estuary is complex, and sediments at the mouth of the estuary exhibit a three-way convergence trend [46]. Sediments near the mouths of the Chanthaburi and Welu rivers in the study area primarily exhibit a two-way migration trend. The sediments in the Pearl River Estuary show a transport trend from sea to land. This seaward transport trend occurs at the mouth. The fine-grained sediments, mainly composed of silt, are distributed in coastal areas [17]. This is consistent with the sediment transport trend and sediment distribution in the Chanthaburi Coastal Area. The surface sediments generally exhibit northwestward and northeastward transport trends from sea to land.

In general, compared with tidal-controlled estuaries in the temperate region of eastern China, terrestrial source sediment in the Chanthaburi and Welu estuaries is not transported far offshore because of the lower sediment contents, runoff amounts, and tidal ranges of the estuaries, but is primarily deposited to the sides or within the estuaries. The surface sediments of the two tropical estuaries in the study area are mainly sandy silt and silty sand, whereas the surface sediments of the Oujiang Estuary and the Pearl River Estuary are relatively fine, primarily consisting of silty clay and clayey silt [17,47]. Sandy silt and clayey silt are widely distributed in the Jiulongjiang Estuary [46], which shows that the hydrodynamic conditions of the tropical estuary in the study area are stronger overall. The surface sediments in the study area mainly show a transport trend from sea to land under

the control of tidal currents; thus, the sediments in most areas are derived from the surrounding sea area. The sediment transport trend of the surface sediments in the Pearl River Estuary is similar to that of the study area, mainly showing transport from sea to land [17]. In contrast, the Jiulongjiang Estuary is affected by the local terrain and complex hydrodynamic conditions, and exhibits correspondingly complex trends in sediment transport [46]. The main reasons for the similarities and differences in the transport trends of sediments in these estuaries are the differences in hydrodynamic conditions and the specifics of regional topography.

## 5. Conclusions

Seven types of surface sediments on the Chanthaburi coast of Thailand were identified. The grain sizes of the study area were mainly sand and silt. The clay content was very small. The sorting was generally poor and gradually became poorer with distance offshore. The skewness was mostly positive. The sand fraction in surface sediments was higher in the mouths of the Chanthaburi and Welu estuaries, whereas the highest silt fractions were distributed near the central part of the study area.

The results of grade-standard deviation analysis indicated that there were two sensitive grain-size components present in the study area: sensitive grain size a (1.25–2$\varphi$) and sensitive grain size b (3.25–4.5$\varphi$). Of these two, sensitive grain size b offered a better indication of environmental change. The high percentage area of sensitive grain size b was mainly distributed in the coastal areas of the Chanthaburi and Welu estuaries and the northwestern part of the study area, indicating that the hydrodynamic changes in these three areas were more significant.

The study area is affected by a variety of hydrodynamic forces such as runoff, tidal currents, and coastal currents. The northwestward coastal currents in winter run through the entire study area, whereas the summer coastal currents flow southeastward to the coast. The sediment presents relatively complex transport trends, mainly characterized by northwestward and northeastward transport from sea to land. The sediments at the mouths of the Chanthaburi Estuary and the Welu River oscillate under the influence of tidal currents.

Based on the results of grade-standard deviation analysis and grain-size trend analysis, the study area was divided into three provinces, representing different sedimentary environments and material sources. Province I is distributed in the nearshore of the Chanthaburi and Welu estuaries. Surface sediments in this area are mainly terrigenous sediments, mixed with small amounts of sediment from the surrounding sea areas. Province II is distributed in the northwestern part of the study area. The surface sediments in this area are relatively coarse sediments remaining in the denuded state. Province III is distributed in the central part of the study area and most of the coastal areas. The fine-grained sediment in the area is derived from the Gulf of Thailand and brought by tidal currents and other processes.

Compared with temperate tidal-controlled estuaries in eastern China, the two tropical estuaries in the study area have smaller suspended sediment contents, runoff amounts, and tidal ranges. Sediment grains are also coarser, indicating that the hydrodynamic conditions are stronger. The surface sediments of the study area mainly show transport trends from sea to land. The main reasons for the similarities and differences in the transport trends of sediments in these estuaries are the differences in hydrodynamic conditions and the specifics of regional topography.

**Author Contributions:** All authors contributed to the data assessment and analysis strategy. C.W. coordinated and wrote the original draft with contributions from the other co-authors. M.C. contributed to the review and editing. H.Q. contributed to the project administration, funding acquisition and paper review. W.I. contributed to the investigation and resources. A.K. contributed to the data curation and data interpretation. All authors have read and agreed to the published version of the manuscript.

**Funding:** This research was funded by the Scientific Research Foundation of the Third Institute of Oceanography, MNR, No. 2019026 and No. 2017034, and the China-ASEAN Maritime Cooperation fund "Monitoring and conservation of the coastal ecosystem in the South China Sea".

**Acknowledgments:** We thank all those who helped to collect samples and data during the survey in Thailand. We thank Guy Evans and Sara J. Mason, from Liwen Bianji, Edanz Editing China (www.liwenbianji.cn/ac), for editing the English text of drafts of this manuscript.

**Conflicts of Interest:** The authors declare no conflict of interest.

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
