# Peer review of "Grain-Size Distribution of Surface Sediments in the Chanthaburi Coast, Thailand and Implications for the Sedimentary Dynamic Environment"

_jmse, doi:10.3390/jmse8040242_

Round 1

Reviewer 1 Report

The authors took into consideration most of my previous comments. Nonetheless, regarding point 7 and sediment transport by saline density currents: authors should include this point in the main text to avoid that a future reader may have the same questio as me. For instance, authors could add the following sentence: Sampling field campaing took place in November, during the dry season of winter monsoon, with low rainfall and low river sediment input. Thereby, most of the sediment transport was conveyed by tidal movement. Density saline currents and its corresponding sediment distal transport [Zordan et al. 2018] was not observed in this location.

Zordan, J., Juez, C., Schleiss, A.J. and Franca, M. J. Entrainment, transport and deposition of sediment by saline gravity currents. Advances in Water Resources. 2018.

Author Response

Thank you for your suggestions.I have took your suggestions and added this explanation to the paper.

Reviewer 2 Report

The paper is reasonably enhanced in regard to the first submission.

I suggest to accept it for publication. However an editing of English language is mandatory.

Author Response

Thank you for your suggestions. Our manuscript has checked by a professional English editing service. The revised paper will be presented in a new manuscript。

This manuscript is a resubmission of an earlier submission. The following is a list of the peer review reports and author responses from that submission.

Round 1

Reviewer 1 Report

Dear Authors,

The topic of your research article is interesting. However, I suggest only small editing to add spaces such as at line 98 and so on or remove of a dot at line 93.

Author Response

Point 1:  Only small editing to add spaces such as at line 98 and so on or remove of a dot at line 93.

Response 1:Thanks,I have added spaces at some appropriate positions.

Reviewer 2 Report

The manuscript analyzes the grain size-distribution of surface sediments depending on the available sediment sources. Statistics applied to the field data collection allows to determine the origin and material source of the sediments. The research herein presented is certainly within the scope of Journal of Marine Science and Engineering.

According to my observations, the topic of the manuscript is interesting and challenging. However, the lack of clarity in some parts of the text should be fixed before the publication. I think the paper requires sharpening in the definition of the results obtained and subsequent discussion. Nonetheless, I am supportive with the manuscript and after the revision herein purposed I think it should be ready for publication. I will be happy to review an updated version of the manuscript.

List of comments

- Authors should remove the following statement in line 14: “…to provide further scientific analysis…”. This statement is rather general and obvious and it thus be deleted.

- Line 24: “The hydrodynamics in the study area are strong”. Please, be more specific on this statement.

- Lines 51-54. In this paragraph (which it is well written and clarifying), I would suggest to add and explain the work of [1] and use their physical insights in the results section text. This work studied the sediment concentration-discharge patterns as a result of the contribution of distal and proximal sediment sources for different grain sizes. These variables are strongly related with the aggradation/degradation processes in alluvial and coastal areas.

- Lines 230-242. Results are explained well in terms of grain size, sorting and skewness of the grain-size curves. However, the linkage with the hydrodynamic forces is not well established: how the trending sediment transport is computed?, quantitatively or based on simulations. All through the manuscript, data related with runoff, tidal currents or coastal currents is not outlined and only in Table 1 (towards the end of the manuscript) some information is provided.

- Lines 255-256. “The interaction forces between runoff, tidal currents and waves are significant.” How did the authors come across with this statement?. This comment needs further clarification.

- Figure 8. How did the authors elaborate such figure?, based on numerical simulations, field data?.

- Lines 298-300. This is an important clarification. I think the authors could link this information with the distal transport of sediments by means of density currents. In [2], it is showed how density currents with different excess of density, are related with the distal sediment transport. Probably, during the wet season (i.e. higher sediment concentration) coarser material is transported farther by means of density currents and ultimately, the summer coastal current may relocate such sediment.  

- I wonder if with the available data, the authors would be able to assess the impact of climate change (if any) in the spatial distribution of the sediments.

Bibliography

[1] The origin of fine sediment determines the observations of suspended sediment fluxes under unsteady flow conditions. C Juez, MA Hassan, MJ Franca. WATER RESOURCES RESEARCH. 2018, 1-16.

[2] Entrainment, transport and deposition of sediment by saline gravity currents. J. Zordan, C. Juez, A. J. Schleiss and M. J. Franca. ADVANCES IN WATER RESOURCES. 2018.

Reviewer 3 Report

Dear authors, 

your manuscript has scientific potential, but unfortunately this is not enough to publish an article. The information you present in the Results section is generally only repeated in the discussion. Please add extensive interpretation and worldwide references. Each claim needs interpretation and worldwide references, rather than just a general description.

Expand and specify the interpretation about the sources. How do you know, that something is a marine source or continental discharge? You sampled the area and you also characterized the samples. Where and which are the differences? Report and write about them.

Please review this manuscript in the above written manner.

Kind regards